# Effect of partner support on antenatal care visits among married adolescents in rural northwestern Uganda: A quasi-experimental study

Saidi Appeli[1]*, Christine Chandia[2,3], Jonathan Izudi[4,5]

1 Department of Agribusiness and Extension, Faculty of Agriculture and Animal Sciences, Busitema University, Soroti, Uganda, 2 Department of Community Health and Behavioral Sciences, School of Public Health, Makerere University, Kampala, Uganda, 3 Programme Office, Restless Development, Arua, Uganda, 4 Directorate of Graduate Training, Research and Innovation, Muni University, Arua, Uganda, 5 Department of Community Health, Mbarara University of Science and Technology, Mbarara, Uganda

* saidiappeli12@gmail.com

## Abstract

### Background

Adolescent pregnancy is associated with several challenges, and partner support is crucial. However, the causal effect of partner support on the use of maternal health services among pregnant adolescents has not been rigorously examined. We assessed the causal effect of partner support on the frequency of antenatal care (ANC) visits among married adolescents in rural Uganda.

### Methods

We conducted a quasi-experimental study using observational data from a cross-sectional study that involved married adolescents aged 10–19 years in rural north-western Uganda. The primary exposure was partner support, measured as a binary variable. Adolescents were considered to have received support if their spouse encouraged ANC attendance, accompanied them to visits, or provided financial or emotional support during ANC; otherwise, they were classified as having not received support. The primary outcome was the number of ANC visits, while the secondary outcome was attending four or more ANC visits. Propensity score weighting was used to ensure covariate comparability between the partner support groups (yes vs. no). Poisson regression was used to estimate the causal effect of partner support on the number of antenatal care (ANC) visits (primary outcome), while the modified Poisson regression was used to estimate the causal effect on attending four or more visits (secondary outcome).

### Results

Of 281 participants, 205 (72.9%) received partner support. Among participants with partner support compared to those without partner support, the frequency of ANC

**Data availability statement:** All relevant data are within the manuscript and its Supporting Information files.

**Funding:** The author(s) received no specific funding for this work.

**Competing interests:** The authors have declared that no competing interests exist.

visits (Risk Ratio 1.15, 95% CI: 1.00–1.32) and four or more ANC visits (Risk Ratio 1.25, 95% CI: 1.01–1.52) improved.

## Conclusion

The study showed that partner support was associated with improvements in ANC visit frequency and attendance of four or more ANC visits, among married adolescents in northwestern Uganda. Interventions aimed at improving ANC utilization should consider engaging and educating partners, as their support positively influences ANC utilization.

## Introduction

Adolescent pregnancy remains a critical public health issue and a risk factor for maternal mortality, particularly in low-resource settings where access to quality antenatal care (ANC) is limited [1–3]. Global estimates indicate that 295,000 women died of pregnancy and childbirth-related complications in 2017 [4]. Moreover, there are 42 births for every 1,000 teenage girls, with the majority of them being in sub-Saharan Africa (SSA) [5]. In Uganda, about 25% of girls aged 15–19 years have begun having children [6]. Adolescence, spanning from 10 to 19 years of age, therefore represents a crucial phase in life characterized by significant biological, psychological, and social transformations [7]. Given the heightened risk of adverse pregnancy outcomes among adolescents resulting from biological immaturity and socio-economic difficulties, early ANC access and partner support [8] are crucial interventions for enhancing health outcomes. Male participation in maternal health can take several forms, such as attending ANC with a spouse, contributing money for ANC events, and offering emotional support, including talking with a partner about problems that arise during ANC [9–12]. In this study, we defined partner support based on adolescents receiving any one of the following: financial or emotional support during ANC, escorting the adolescent to ANC visits, or encouraging ANC attendance from the partner/husband. Male participation in ANC is linked to improved access to skilled birth attendance, birth preparedness and complication readiness, including fewer delays in seeking medical attention when needed [9,12,13].

Several studies have examined partner involvement in maternal health [2,12,14,15]. For instance, a systematic review and meta-analysis of male involvement and maternal health outcomes showed that partner involvement improved the overall utilization of maternal health services [15]. In a study conducted in Myanmar to examine the association between partner involvement and maternal utilization of antenatal, intrapartum (delivery), and postnatal services, findings showed that women who were not accompanied to attend ANC more than once by their partner were less likely to attend more than four ANC visits [16]. In Afghanistan, pregnant women who were not accompanied to ANC by their partners had a lower likelihood of adequate ANC utilization, including commencing ANC visits during the first trimester [2].

Although partner support is associated with ANC utilization in previous studies [14,17,18], it is uncertain whether this relationship is causal. The past studies performed multivariable analysis (covariate adjustment), which not only assumes no unmeasured confounding but is also susceptible to model misspecification.

Therefore, the evidence around the true effect of partner support on the use of maternal health services such as ANC among pregnant adolescents in general, and the northwestern Uganda in particular, remains uncertain. A lack of partner support may result in under-utilization of existing health services due to factors such as inadequate financial resources and psychological stress, which may make adolescents reluctant to seek ANC care. We, therefore, examined the causal effect of partner support on the frequency of ANC visits and attendance of ≥4 ANC visits among married adolescents in rural northwestern Uganda. This evidence may provide the basis for advocating for partner support among pregnant adolescents in Uganda and beyond.

## Materials and methods

### Study design

We designed a quasi-experimental study from existing observational data, as randomization was not possible [19,20]. The exposure preceded the outcome, so cause-effect analysis was possible. The exposed group consisted of adolescents who received partner support, while the unexposed group consisted of adolescents without the support. The two groups differed in many ways, including both measured and unmeasured characteristics, as the groups were not randomly assigned. Therefore, to emulate a randomized experiment, we used propensity score weighting to remove systematic differences between the groups by weighting the groups using propensity scores. This approach balances the groups (exposed and unexposed) by giving them weights based on their likelihood of being exposed, considering their characteristics [21]. By doing this, we reduced the bias and made the groups more comparable, similar to a randomized trial. The propensity score is simply the chance (from 0 to 1) that an adolescent would be in the exposed group, based on their observed characteristics [22,23].

### Data source and study setting

The data (S1 File) analyzed in this study were from a parent study, a cross-sectional study conducted among pregnant adolescents to examine their utilization of ANC services in Ayivu West division in Arua City, West Nile, Uganda. The West Nile region borders the Democratic Republic of Congo (DRC) to the west and had an estimated population of 3,316,255 in 2024. Arua City was estimated to have 384,656 residents, of whom 26,770 were adolescent girls aged 10–19 years. During the first and second quarters of the 2023/2024 financial year, 1,172 adolescents became pregnant and sought ANC services across health facilities in Ayivu West [24]. For this reason, the parent study was conducted to understand the determinants of pregnancy among adolescents aged 10–19 years. The parent study used a sample size of 409 participants, estimated using William Cochran's formula [25]. The study assumed a 41.2% proportion of adolescents as utilizing ANC services in Arua City [26], a 95% confidence level (Z = 1.96) with a 5% margin of error (e = 0.05), and a 10% non-response rate. Cluster sampling (adolescents from three selected sub-counties of Pajulu, Adumi, and Ayivu) and simple random sampling techniques were used to guarantee unbiased sample participant selection. Structured questionnaires were used to collect data through KoboCollect version 2024.2.4. The data collection period was from October 28, 2024, to January 24, 2025. During data collection, participants were provided with a detailed explanation of the study objectives, procedures, potential benefits, risks, and their right to voluntary participation, including the freedom to withdraw at any stage without consequences. Recognizing the potential vulnerability of adolescents and the sensitive nature of the study topic, research assistants first established trust by assuring participants of strict confidentiality and privacy throughout the research process.

This approach fostered open and honest discussions, allowing participants to express themselves freely without fear of judgment or disclosure of personal information. Additionally, all research assistants signed confidentiality agreements to

 

formally commit to maintaining the privacy and confidentiality of all collected data. The data collected included socio-demographic characteristics and ANC utilization, including partner support. Participants were asked to report the number of ANC visits they had received during their most recent pregnancy. Additionally, the participants were asked to state whether they had received any support from their partner (partner support). The survey included specific questions on whether the husband had encouraged her to attend ANC, accompanied her to ANC, and provided financial or emotional support. For the present study, our analysis focused on married adolescents aged 10–19 years, resulting in a final sample size of 281 complete observations. All participants were at least 36 weeks of gestation and had no complications; therefore, they were not anticipated to require additional ANC visits. We excluded unmarried pregnant adolescents or those without ANC cards, as we could not verify ANC visits. As ethical approval was obtained during the parent study, no additional ethical approval was needed for re-analyzing the existing data.

## Variables and measurements

**Outcome.**  The primary outcome was the number of antenatal care (ANC) visits, measured as a count data. Based on local ANC guidelines, pregnant women are expected to attend ≥4 ANC visits during pregnancy. To reflect the recommendation, we categorized ANC visits as <4 vs. ≥ 4 to form the secondary outcome as an utilization proxy, not a quality or timeliness indicator.

**Exposure (intervention) variable.**  The exposure (intervention) variable was partner support, categorized as no vs. yes. An adolescent was considered to have received partner support if her spouse supported her in any one of the following domains: 1) encouraging ANC attendance, 2) accompanying her to visits, and 3) providing financial or emotional support during ANC. The exposed group consisted of adolescents with partner support, while the unexposed group comprised those without the support.

**Covariates.**  We included the following baseline covariates: age as a categorical variable (10–14 versus 15–19 years), religion (Catholic, Muslim, Pentecostal/adventist, and other), residence (peri-urban versus rural), level of education (no formal education, primary education, secondary, and tertiary), parity (1–2 versus 3–4), type of occupation (farming, selling goods at the market, working for someone, other), time to reach the health facility for ANC (<1 hour, 1–2 hours, ≥ 3 hours), healthcare provider attitude (friendly or unfriendly), and whether or not counseling services were received. These factors were chosen based on their availability in the parent study and evidence from the literature. The categorization of parity signifies mothers with limited versus greater experience in utilizing maternal health services, including ANC, skilled birth attendance, and postnatal care. Categorizing travel time as <1 hour, 1–2 hours, and ≥3 hours enables assessment of how increasing distance affects ANC use.

## Statistical analysis

The overall statistical analysis was done using Stata/SE 15.0 (S2 File). We descriptively summarized categorical covariates using frequencies and percentages. We cross-tabulated the covariates by partner support status (no vs. yes) and assessed differences in covariate distribution using tests of statistical significance at a 5% level. The Chi-square test was used to assess differences in proportions between categorical variables and partner support when cell counts were ≥5; otherwise, Fisher's exact test was used. Using the Wilcoxon rank-sum test, we assessed the median distribution of the number of ANC visits among adolescents, comparing those with and without partner support. We grouped the variables based on partner support (no vs. yes) and checked if the two groups were similar by comparing them using absolute standardized mean difference (SMD). If the absolute standardized mean difference (SMD) was less than 0.1, we considered the groups balanced; else, it was unbalanced [22,27]. We fitted a multivariable logistic regression model as a function of the exposure and the covariates and used the estimated coefficients to predict propensity scores as described previously [28–30]. The propensity score is the probability of being in the exposed group conditional on the observed participant's characteristics [22]. We assessed the correctness of the propensity score model using the Hosmer-Lemeshow test. Our

null hypothesis was that the propensity score model was correctly specified [31]. To make sure the two groups were similar, we used inverse probability weighting (IPW) to adjust the weights based on the propensity score. The exposed group was weighted using the inverse of the propensity score (1/propensity score), while the unexposed group was weighted using the inverse of one minus the propensity score (1/(1- propensity score)) [32]. We performed propensity score weighting using the "*propwt*" command in Stata from a community-contributed package [31]. Then, we checked if the groups were balanced by comparing their characteristics, with absolute SMD < 0.1 considered confirmatory of covariate balance [33,34].

After achieving covariate balance, to examine whether partner participation directly influences ANC utilization (causal analysis), a propensity-score weighted Poisson regression model for count data, adjusted for propensity score weights, was employed to examine the causal effect of partner support on the primary outcome. Findings are reported using the risk ratios (RR) and 95% confidence interval (CI). To measure the causal effect of partner support on the secondary outcome, the propensity score weighted modified poisson regression model for binary outcome adjusted for propensity score weights was used.

The result was reported using risk ratios (RR) and 95% CI. A non-causal analysis using unadjusted and adjusted Poisson regression for the primary outcome and modified poisson regression for the secondary outcome was performed to supplement the causal analysis results and assess if the direction of evidence remained the same or converged.

We assessed whether or not the functional form of the logistic regression model used to estimate the propensity score was appropriate. This was done using the Hosmer-Lemeshow test, which evaluates the goodness of fit of the model. The Hosmer-Lemeshow test (chi2(8) = 9.27, Prob > chi2 = 0.320) results indicated no significant deviation, suggesting that the propensity score model was correctly specified. We assessed the Poisson regression assumption of equidispersion using the Pearson-based dispersion statistic. Over-dispersion is suggested when the dispersion statistic exceeds one. In our analysis, the Pearson dispersion statistic (Pearson $\chi^2$/df = 0.582) was well below this threshold, indicating no evidence of substantial over-dispersion in the data. The Poisson goodness-of-fit test for the adjusted model shows no significant lack of fit, with deviance (210.99, p = 0.987) and Pearson (208.08, p = 0.991) statistics.

## Ethical considerations

This study involved the analysis of existing data from a previous study. The parent study was approved by the Makerere University School of Public Health Research and Ethics Committee (MakSPH-REC), and the ethical approval number is MakSPH-REC 530. The participants gave written informed consent. No additional ethical approval was required for this study, as it involved analysis of existing data for which participants had provided written informed consent

## Results

### Characteristics of participants

We analyzed data on 281 married adolescents: 205 with partner support vs. 76 without partner support (Table 1). The majority of the adolescents were aged 15–19 years (98.2%), identified themselves as Catholic (73.3%), and lived in rural areas (64.4%). Among the Catholics, 76.6% had a supportive partner, among Muslims, 6.8% and among Pentecostal/Adventists, 8.3%. A majority (70.7%) of adolescents with partner support resided in rural areas, while 29.3% lived in peri-urban areas. Adolescents with partner support were less likely to reach a health facility for ANC within one hour (35.2%) compared to those without partner support (54.0%). Furthermore, 66.8% of adolescents with partner support demonstrated adequate ANC utilization compared with 51.3% of adolescents without partner support. No significant difference was observed in the following covariates, namely age, level of education, number of children (parity), nature of occupation, healthcare provider attitude, and whether or not the adolescent received counseling services at the health facility (all p > 0.05).

**Table 1. Characteristics of participants.**

| Variables | Level | Overall, (%) | Partner support | | P-value |
|---|---|---|---|---|---|
| | | | No, n(%) | Yes, n(%) | |
| | Sample sizes | (n = 281, 100%) | (n = 76, 27.1%) | (n = 205, 72.9%) | |
| Age (years) | 10-14 | 5 (1.8) | 2 (2.6) | 3 (1.5) | 0.615 |
| | 15-19 | 276 (98.2) | 74 (97.4) | 202 (98.5) | |
| Religion | Catholic | 206 (73.3) | 49 (64.5) | 157 (76.6) | 0.010 |
| | Muslim | 21 (7.5) | 7 (9.2) | 14 (6.8) | |
| | Pentecostal/Adventist | 34 (12.1) | 17 (22.4) | 17 (8.3) | |
| | Other | 20 (7.1) | 3 (3.9) | 17 (8.3) | |
| Residence | Peri-urban | 100 (35.6) | 40 (52.6) | 60 (29.3) | <0.001 |
| | Rural | 181 (64.4) | 36 (47.4) | 145 (70.7) | |
| Level of education | No education | 27 (9.6) | 9 (11.8) | 18 (8.8) | 0.068 |
| | Primary | 212 (75.4) | 50 (65.8) | 162 (79.0) | |
| | Secondary | 37 (13.2) | 14 (18.4) | 23 (11.2) | |
| | Tertiary | 5 (1.8) | 3 (4.0) | 2 (1.0) | |
| Parity | 1–2 | 272 (96.8) | 74 (97.4) | 198 (96.6) | 1.000 |
| | 3–4 | 9 (3.2) | 2 (2.6) | 7 (3.4) | |
| Nature of occupation | Farming | 130 (46.3) | 26 (34.2) | 104 (50.7) | 0.056 |
| | Selling goods at the market | 74 (26.3) | 23 (30.3) | 51 (24.9) | |
| | Working for someone | 23 (8.2) | 10 (13.2) | 13 (6.3) | |
| | Other | 54 (19.2) | 17 (22.3) | 37 (18.1) | |
| Time to the health facility for ANC (hours) | Less than 1 | 113 (40.2) | 41 (54.0) | 72 (35.2) | 0.001 |
| | 1- 2 | 136 (48.4) | 33 (43.4) | 103 (50.2) | |
| | ≥3 | 32 (11.4) | 2 (2.6) | 30 (14.6) | |
| Healthcare provider attitude | Friendly | 261 (92.9) | 73 (96.1) | 188 (91.7) | 0.297 |
| | Unfriendly | 20 (7.1) | 3 (3.9) | 17 (8.3) | |
| Received counseling services at the health facility | No | 103 (36.7) | 32 (42.1) | 71 (34.6) | 0.248 |
| | Yes | 178 (63.3) | 44 (57.9) | 134 (65.4) | |
| At least four ANC visits | No | 105 (37.4) | 37 (48.7) | 68 (33.2) | 0.017 |
| | Yes | 176 (62.6) | 39 (51.3) | 137 (66.8) | |
| Number of ANC visits | Median (IQR) | 4.0 (5−3) | 3.5 (5−3) | 4.0 (5−3) | 0.036 |

## Distribution of the outcomes for adolescents comparing those with and without partner support

Table 2 summarizes the distribution of the outcomes comparing participants with and without partner support. More participants with partner support (66.8%) utilized ANC services compared to those without partner support (51.3%). The median number of ANC visits was higher among participants with partner support compared with those without partner support.

## Covariate balance before and after propensity-score weighting

Table 3 shows covariate balance before and after propensity-score weighting between participants with vs. without partner support. Before propensity score weighting, several covariates were not balanced between the two groups, as the absolute SMD was more than 0.1. After applying propensity-score weighting, covariate distributions between the two groups were well-balanced, with all SMDs below 0.1. The covariate balance between the two groups implies that the two groups were similar; hence, confounding and selection bias had been removed through propensity score weighting.

**Table 2. Distribution of the outcomes for adolescents with and without partner support.**

| Outcomes | Levels | All (n = 281) | Partner support | | P-value |
|---|---|---|---|---|---|
| | | | No (n = 76) | Yes (n = 205) | |
| At least four ANC visits | No | 105 (37.4) | 37 (48.7) | 68 (33.2) | 0.017 |
| | Yes | 176 (62.6) | 39 (51.3) | 137 (66.8) | |
| Number of ANC visits | Median (IQR) | 4.0 (5−3) | 3.5 (5−3) | 4.0 (5−3) | 0.036 |

**Table 3. Covariate balance before and after propensity-score weighting.**

| | Covariate distribution before propensity-score weighting (n = 281) | | | Covariate distribution after propensity-score weighting (n = 281) | | |
|---|---|---|---|---|---|---|
| | Partner support | | | Partner support | | |
| | No | Yes | | No | Yes | |
| Variables | Mean (SD) | Mean (SD) | SMD | Mean (SD) | Mean (SD) | SMD |
| Age (years) | 1.97 (0.16) | 1.99 (0.12) | 0.082 | 1.98 (0.13) | 1.98 (0.13) | −0.009 |
| Religion | 1.66 (0.96) | 1.48 (0.96) | −0.182 | 1.54 (0.94) | 1.54 (0.99) | 0.000 |
| Residence | 1.47 (0.50) | 1.71 (0.46) | 0.487 | 1.68 (0.47) | 1.65 (0.48) | −0.052 |
| Level of education | 2.14 (0.67) | 2.04 (0.49) | −0.172 | 2.06 (0.64) | 2.07 (0.50) | 0.013 |
| Parity | 1.03 (0.16) | 1.03 (0.18) | 0.046 | 1.04 (0.19) | 1.03 (0.18) | −0.037 |
| Nature of occupation | 2.24 (1.15) | 1.92 (1.14) | −0.279 | 1.95 (1.16) | 1.99 (1.16) | 0.035 |
| Time to the health facility for ANC | 2.11 (0.99) | 1.85 (0.91) | −0.270 | 1.87 (0.98) | 1.91 (0.93) | 0.040 |
| Healthcare provider attitude | 1.04 (0.20) | 1.08 (0.28) | 0.181 | 1.05 (0.21) | 1.07 (0.26) | 0.081 |
| Received counseling services at the health facility | 1.58 (0.50) | 1.65 (0.48) | 0.153 | 1.66 (0.48) | 1.64 (0.48) | −0.036 |

Abbreviations: SMD-Absolute standardized mean difference: SD-standard deviation

### Effect of partner support on the frequency of ANC visits before and after propensity-score weighted analysis

In the causal analysis (Table 4), partner support tended to improve the frequency of ANC visits (adjusted RR = 1.15, 95% CI:1.00–1.32). In the non-causal analysis, the improvement in the frequency of ANC visits was 22% in the adjusted analysis (aRR = 1.22, 95% CI:1.02–1.46) and 33% in the unadjusted analysis (unadjusted RR = 1.33, 95% CI:1.10–1.60).

### Effect of partner support on attendance of four or more ANC visits before and after propensity-score weighted analysis

Following causal analysis (Table 4), the results showed that partner support significantly increased attendance of four or more ANC visits by 25% (adjusted RR = 1.25, 95% CI: 1.01–1.52). The non-causal analysis showed similar results in the adjusted (adjusted RR = 1.28, 95% CI: 1.00–1.65) and unadjusted analysis (unadjusted RR = 1.30, 95% CI: 1.02–1.66).

## Discussion

We evaluated the causal effect of partner support on the frequency of ANC visits as well as ≥4 ANC visits among married adolescents in rural Uganda. We found that partner support improved the number of ANC visits and the proportion of individuals attending four or more of the visits. Our findings align with those of a mixed-methods study in rural Southwestern Nigeria, which found that partner support during pregnancy positively influenced women's experiences during pregnancy, labor, and delivery [35]. Also, in Northwestern Tanzania, partner involvement in ANC showed a significant association with four or more ANC visits [36]. Similarly, in southern Ethiopia, women who received partner support had a higher proportion

**Table 4. Effect of partner support on ANC visits.**

| Outcome | | Causal analysis | Non-causal analysis | |
|---|---|---|---|---|
| | | Propensity-score weighted analysis | Adjusted regression analysis | Unadjusted regression analysis |
| | | aRR (95% CI) | aRR (95% CI) | Unadjusted RR (95% CI) |
| *A. Number of ANC visits* | Partner support | | | |
| | No | 1 | 1 | 1 |
| | Yes | 1.15* (1.00-1.32) | 1.22** (1.02-1.46) | 1.33** (1.10-1.60) |
| *B. ≥ 4 ANC visits* | Partner support | | | |
| | No | 1 | 1 | 1 |
| | Yes | 1.25* (1.01-1.52) | 1.28* (1.00-1.65) | 1.30* (1.02-1.66) |

**Note**: Risk ratio (RR) are the exponentiated beta coefficients; 95% confidence intervals in brackets. The Poisson regression model was used for the analysis of the number of ANC visits. Modified poisson regression analysis was used for ≥4 ANC visits. aRR: Adjusted risk ratio; RR: Unadjusted risk ratio. Significance; *$p < 0.05$, **$p < 0.01$, *** $p < 0.001$. Adjusted analysis included the following covariates: age (years), religion, residence, level of education, parity, nature of occupation, time to the health facility for ANC health care provider attitude, and receiving counseling services at the health facility.

of ANC attendance compared to those without the support [14]. However, unlike our study, the findings are from a descriptive analysis. Our findings, therefore, provide a less-biased estimate of the effect of partner support on ANC visits as a rigorous analytic approach was employed to address selection bias and confounding.

Several factors may have contributed to men giving partner support. For example, partner support may be influenced by family and gender roles that underscore the partner's responsibility for the well-being of the wife during pregnancy. Additionally, some men may receive health education from healthcare providers, non-governmental organizations, or traditional birth attendants, leading them to recognize that regular ANC visits can prevent complications during childbirth, including reducing risks for both the mother and newborn baby. Accordingly, the study findings underscore the importance of partner involvement as a modifiable factor that may enhance maternal use of ANC. It is, therefore, imperative for interventions aimed at increasing ANC utilization to consider incorporating strategies to engage and educate partners, as their support improves maternal health services-seeking behaviors [37,38].

## Strengths and limitations of this study

Our study has several strengths. We used a strong causal analysis method to establish an unbiased causal effect of partner support on ANC. We correctly specified the propensity score model, hence strengthening the credibility of the causal estimates. Both causal and non-causal findings showed convergence of causal effect estimates, suggesting that the findings are robust. Therefore, our findings most likely will hold even when considering potential hidden biases or unmeasured factors.

Additionally, to the best of our knowledge, this is the first study in rural northwestern Uganda to evaluate the causal effect of partner support on the frequency of ANC visits among married adolescents. Our study acknowledges some limitation. The relatively small sample size supported causal estimation within the study population but limits the generalizability of the findings to other settings or populations with different characteristics. The study relied on self-reported data regarding partner support, which may be subject to response bias, including potential ambiguity in the timing of exposure relative to ANC attendance. Also, effects should be interpreted cautiously as there may be a possibility of over-estimation of effective ANC utilization since the visits did not consider timing [39]. Additionally, the specific domains of partner support, including whether the husband had encouraged her partner to attend ANC, accompanied her to ANC, and provided financial or emotional support, with potentially varying effects on the outcome, were not studied. Some categorical

variables in the previous study may not have been optimally categorized, and reclassification was not possible due to data limitations. For instance, travel time could have been more appropriately categorized as <1 hour, 1–2 hours, 3–4 hours, and >4 hours to better capture differences in access to health facilities. Also, the dataset was missing important covariates that relate to the education level of the partner and pregnancy intention. This study was restricted to married adolescents, which allowed for reliable measurement of partner support but limits the generalizability of the findings to all adolescents. Unmarried adolescents may experience different levels or types of partner support, and including them could have introduced misclassification of exposure. Consequently, conclusions regarding partner support should be interpreted within the context of married adolescents. While adjustments were made for confounding variables using propensity-score weighting, unmeasured confounders may not have been completely removed and possibly influence the results. Lastly, ANC attendance was defined as fewer than four versus at least four visits based on the available data. This measure captures the number of ANC contacts but not their timing, which may overestimate timely or adequate ANC utilisation. Therefore, our findings should be interpreted as reflecting attendance of at least four ANC visits rather than adherence to recommended timely ANC schedules. Future studies with detailed information on visit timing are needed to better align measurement with current WHO recommendations.

## Conclusion

Among pregnant adolescents in northwestern Uganda, this study showed that partner support increased ANC visits, both the frequency and attendance of four or more visits. This suggests that interventions aimed at encouraging partner involvement in maternal healthcare could help improve maternal and neonatal health outcomes in this setting. Future investigation is needed to explore barriers to partner support in promoting maternal healthcare utilization.

## Supporting information

**S1 File. Dataset.**
(CSV)

**S 2 File. Statistical codes.**
(DOCX)

## Acknowledgments

We thank the research assistants for their dedication and time devoted to the parent study.

## Author contributions

**Conceptualization:** Saidi Appeli, Christine Chandia, Jonathan Izudi.

**Data curation:** Saidi Appeli, Christine Chandia.

**Formal analysis:** Saidi Appeli, Jonathan Izudi.

**Investigation:** Saidi Appeli, Christine Chandia, Jonathan Izudi.

**Methodology:** Saidi Appeli, Jonathan Izudi.

**Project administration:** Christine Chandia.

**Resources:** Christine Chandia.

**Software:** Saidi Appeli.

**Supervision:** Jonathan Izudi.

**Validation:** Saidi Appeli, Jonathan Izudi.

**Visualization:** Saidi Appeli, Jonathan Izudi.

**Writing – original draft:** Saidi Appeli, Christine Chandia, Jonathan Izudi.

**Writing – review & editing:** Saidi Appeli, Christine Chandia, Jonathan Izudi.

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
