## [Decision Letter · Decision Letter 0]

5 Nov 2025

Dear Dr. Appeli,

Thank you for submitting your manuscript to PLOS ONE. After careful consideration, we feel that it has merit but does not fully meet PLOS ONE’s publication criteria as it currently stands. Therefore, we invite you to submit a revised version of the manuscript that addresses the points raised during the review process.

 Could you please carefully revise the manuscript to address all comments raised? 

We look forward to receiving your revised manuscript.

Kind regards,

Helen Howard

Staff Editor

PLOS ONE

Journal Requirements:

3. We note that your Data Availability Statement is currently as follows: All relevant data are within the manuscript and its Supporting Information files

Reviewers' comments:

Reviewer's Responses to Questions

**Comments to the Author**

1. Is the manuscript technically sound, and do the data support the conclusions?

Reviewer #1: Yes

2. Has the statistical analysis been performed appropriately and rigorously?

Reviewer #1: Yes

3. Have the authors made all data underlying the findings in their manuscript fully available?

Reviewer #1: Yes

4. Is the manuscript presented in an intelligible fashion and written in standard English?

Reviewer #1: Yes

Reviewer #1: Overall

This is a relevant article based on thorough analyses of data.

Introduction

Introduction doesn't always flow well. It can use some improvement.

The sentence starting in line 57 does flow well, specifically the part "including the role of the partner [7],".

The sentence starting in line 62 needs the addition of "from the partner/husband" after the following. "By the partner" at the end of the sentence can then be removed.

The word "Moreover" at the beginning of the sentence starting at line 64 can be removed.

In the paragraph starting at line 76, the inadequacy of cross-sectional study designs is discussed. This remark sounds odd, given that the current study is also using cross-sectional data. You are grading down your study, before it has even started. Later on it becomes clear that the data analysis technique corrects for this, but perhaps other studies with a cross-sectional have done the same. Then it is suggested that the cross-sectional design is not the restriction in this study.

Method

In line 95 it is mentioned that there are 1,172 teenage pregnancies per annum. To judge for themselves if this a high prevalence, the reader needs to know among how many pregnancies in total this is.

In line 99, add participants after 409.

How were the participants approached? What kind of relationship was there between the data collectors and the participants?

Line 149 says 19 to 14, I assume this the 19 must be 10.

Line 152: the cut-off for parity is 3 children. What is the rational for this? The reader will be interested in knowing what it is like, with a first child compare to more children.

Line 153 indicates time to reach a health facility for ANC. It is more logical to use <1 hour, 1-2 hours, 3-4 hours and > 4 hours.

Results

Line 194 says "A higher proportion of Catholic adolescents had supportive partners (76.6%) compared to their Muslim (6.8%) and Pentecostal/Adventist (8.3%) counterparts." This is logical as Catholics are by far the largest group. What is of more interest to the reader is that among the Catholics 76.2% had a supportive partner, among Muslims 66.6.% and among Pentecostal 50%. (Of course even then these are unadjusted findings.)

Line 199 mentions "Furthermore, 66.8% of adolescents with partner support demonstrated adequate ANC utilization." How is this for adolescents without partner support?

Table 2: the numbers for yes and no in partner support are switched around.

Line 237: the term "causal analysis" is used here for the first time. The term is not, but should be, explained in the method section.

Discussion

Line 267. The word "explain" is not adequate here. It needs to be replaced by "... may have contributed to men giving partner support." This is what you are exploring here.

Line 274: I miss reference to articles that already explored this and suggestions for further research how this can be done.

Line 282: "hold" should be replaced by "... most likely will hold...".

Line 292: the end of the sentence needs adding something like "...and possibly influence the results." That is why you can only say most likely will hold...

.

Reviewer #1: No

---

## [Author Response · Author response to Decision Letter 1]

10 Nov 2025

We thank you and the reviewer for the valuable comments which have helped to improve the quality of the manuscript. We have addressed all the reviewer comments and indicated in the response letter how each comment was addressed by replicating the text in the manuscript.

---

## [Decision Letter · Decision Letter 1]

13 Jan 2026

Dear Dr. Saidi Appeli

Thank you for submitting your manuscript to PLOS ONE. After careful consideration, we feel that it has merit but does not fully meet PLOS ONE’s publication criteria as it currently stands. Therefore, we invite you to submit a revised version of the manuscript that addresses the points raised during the review process.

We look forward to receiving your revised manuscript.

Kind regards,

Nigus Kassie Worku, Lecturer

Academic Editor

PLOS One

Journal Requirements:

Additional Editor Comments:

The manuscript explores an important and timely topic—partner support and antenatal care utilization among married adolescents in rural Uganda. The study is well-conceived, and the use of quasi-experimental design with propensity score weighting strengthens the analytical rigor. The findings are relevant for maternal health policy and highlight practical implications for engaging male partners in ANC programs.

Overall, the manuscript is clearly structured, and the results are presented logically. However, improvements in language clarity, grammar, and conciseness would enhance readability. The Methods section could provide clearer justification for variable selection and categorization, and some statistical interpretations (e.g., borderline significance) should be stated cautiously. Strengthening the discussion of limitations, contextual factors, and the implications of residual confounding would further improve the manuscript.

In summary, the study is valuable, methodologically sound, and contributes to the literature on adolescent maternal health. With minor revisions to language, clarity, and statistical presentation

Reviewers' comments:

Reviewer's Responses to Questions

**Comments to the Author**

Reviewer #1: All comments have been addressed

Reviewer #2: (No Response)

Reviewer #3: (No Response)

Reviewer #4: All comments have been addressed

2. Is the manuscript technically sound, and do the data support the conclusions?

Reviewer #1: Yes

Reviewer #2: Yes

Reviewer #3: Yes

Reviewer #4: Yes

3. Has the statistical analysis been performed appropriately and rigorously?

Reviewer #1: Yes

Reviewer #2: Yes

Reviewer #3: Yes

Reviewer #4: I Don't Know

4. Have the authors made all data underlying the findings in their manuscript fully available?

Reviewer #1: Yes

Reviewer #2: Yes

Reviewer #3: Yes

Reviewer #4: Yes

5. Is the manuscript presented in an intelligible fashion and written in standard English?

Reviewer #1: Yes

Reviewer #2: Yes

Reviewer #3: Yes

Reviewer #4: Yes

Reviewer #1: (No Response)

Reviewer #2: Effect of partner support on antenatal care visits among married adolescents in rural north-western Uganda: a quasi-experimental study

Abstract

1. Why are the authors reporting RR and OR?

2. Just the at least 4 ANC visits may not really determine the women received the quality services. I suggest that the authors read these two papers to relook and/or refocus their analysis: https://journals.plos.org/plosone/article?id=10.1371/journal.pone.0263650 and https://mjz.co.zm/index.php/mjz/article/view/775

Introduction

3. Line 84: A lack of partner support may result in non-utilization… consider changing this from non-utilisation to under-utilisation to imply general lower uptake than expected.

Methods

4. Start with study design then describe the data sources and variables then analysis then the ethical considerations

5. For the outcome variables, consider reviewing them in line with the two papers provided above so that issues of quality and reduction of over-estimation of ANC utilisation is curbed.

Results

6. Diagnosis for propensity score and Poisson regression model is more of the methods and should be reported under methods.

7. Table 4 should include the other variables as well. The variables that you did the adjustment for the effect that you are presenting.

8. All data are presented in tables, consider using graphs like forest plot to show the uptake of timely ANC visits by the socio-demographic characteristics?

9. May be another forest plot for the adjusted OR for the final model.

Discussion

10. Discussion has not been reviewed since the definition of the 4 ANC visits should be timely and not just at least 4 visits based on the two papers that have found over-estimation if the visits are lumped.

Reviewer #3: Statistical methods use are sound, however, authors need to provide clarity on the choice of these methods. More comments are in attached file

Reviewer #4: I am concerned about the selection of adolescents that were married. Is it not likely that adolescents that were married are likely to be the ones accepted by their spouses than those that were not married, Therefore, the married ones were also more likely to receive spousal support. This should be discussed and highlighted as a limitation. Therefore, the final conclusions should be made in light of this limitation.

.

Reviewer #1: No

Reviewer #2: No

Reviewer #3: **Yes:** Edson MwebesaEdson MwebesaEdson MwebesaEdson Mwebesa

Reviewer #4: **Yes:** Mangwi Ayiasi RichardMangwi Ayiasi RichardMangwi Ayiasi RichardMangwi Ayiasi Richard

---

## [Author Response · Author response to Decision Letter 2]

26 Jan 2026

Response to reviewer comments has been attached

---

## [Decision Letter · Decision Letter 2]

15 Mar 2026

Effect of partner support on antenatal care visits among married adolescents in rural northwestern Uganda: a quasi-experimental study

PONE-D-25-35574R2

Dear Dr. Appeli,

We’re pleased to inform you that your manuscript has been judged scientifically suitable for publication and will be formally accepted for publication once it meets all outstanding technical requirements.

Kind regards,

José Antonio Ortega, Ph.D.

Academic Editor

PLOS One

Additional Editor Comments (optional):

Three among the previous referees were satisfied with the revision and recommend acceptance. The fourth referee was not available but the academic editor has verified that the comments were addressed.

Reviewers' comments:

Reviewer's Responses to Questions

**Comments to the Author**

Reviewer #1: All comments have been addressed

Reviewer #3: All comments have been addressed

Reviewer #4: All comments have been addressed

2. Is the manuscript technically sound, and do the data support the conclusions?

Reviewer #1: Yes

Reviewer #3: Yes

Reviewer #4: Yes

3. Has the statistical analysis been performed appropriately and rigorously?

Reviewer #1: I Don't Know

Reviewer #3: Yes

Reviewer #4: Yes

4. Have the authors made all data underlying the findings in their manuscript fully available?

Reviewer #1: Yes

Reviewer #3: Yes

Reviewer #4: Yes

5. Is the manuscript presented in an intelligible fashion and written in standard English?

Reviewer #1: Yes

Reviewer #3: Yes

Reviewer #4: Yes

Reviewer #1: Overall, the comments have been well addressed and the article has been well adjusted. It reads well.

Reviewer #3: Accept for publication. Authors have addressed all the comments. Congratulations to the authors on this achievement.

Reviewer #4: (No Response)

.

Reviewer #1: No

Reviewer #3: **Yes:** Edson MwebesaEdson MwebesaEdson MwebesaEdson Mwebesa

Reviewer #4: **Yes:** Richard Mangwi AyiasiRichard Mangwi AyiasiRichard Mangwi AyiasiRichard Mangwi Ayiasi

---

## [Editor Report · Acceptance letter]

PONE-D-25-35574R2

PLOS One

Dear Dr. Appeli,

I'm pleased to inform you that your manuscript has been deemed suitable for publication in PLOS One. Congratulations! Your manuscript is now being handed over to our production team.

Kind regards,

on behalf of

Dr. José Antonio Ortega

Academic Editor

PLOS One